# DECOMPOSE TIME AND FREQUENCY DEPENDENCIES: MULTIVARIATE TIME SERIES PHYSIOLOGICAL SIGNAL EMOTION RECOGNITION

## ABSTRACT

In this study, we proposed a transformer based end-to-end solution to capture the relationship between the physiological signals and affective changes. We first convert the physiological signal emotion recognition prediction task to a sequence-to-sequence multivariate time series prediction task. We utilize the state-of-the-art (SOTA) self-attention mechanism to decompose the physiological signals into separate frequency domain and time domain representations and capture the channel dependencies via Two-Stage Attention (TSA). Meanwhile, we implement the multitask learning framework to better predict the valence and arousal affective states individually. We evaluate our system on the Continuously Annotated Signals of Emotion (CASE) dataset used in the Emotion Physiology and Experience Collaboration (EPiC) challenge, and our proposed system outperform all the challenge participants in all four test scenarios.

## 1 INTRODUCTION

Affective Computing is defined by Rosalind Picard as computation that relates to, arises from, or deliberately influences emotions. Picard (1997) The computation can be driven by audio, visual, textual, physiological, and multimodal inputs and the annotation of the task can be categorical (e.g., 7-8 basic emotion), discrete (e.g., discrete Valence/Arousal(VA) space), and continuous (e.g., continuous VA space). While emotion recognition with audio, visual and textual inputs has been benefited from the development of auto-speech recognition(ASR), Computer Vision(CV) and Natural Language Processing(NLP),Chen et al. (2023)Ma et al. (2021)Acheampong et al. (2021), state-of-the-art(SOTA) results of emotion recognition with physiological signal is still dominated by feature engineering and machine learning model.Bota et al. (2019) Until recently, several studies have investigated transformer based solution of physiological emotion recognition. Vu et al. (2023); Vazquez-Rodriguez et al. (2022b;a); Yang et al. (2022) Inspired from the success of theses studies we proposed a transformer based multivariate time-series model to solve the physiological emotion recognition problem with continuous valence and arousal emotion states annotation. We apply the recent SOTA model in time series prediction task to physiological emotion recognition.

Information in both the time domain and the frequency domain is significant for physilogical emotion recognition. Therefore, we choose to use wavelet and Fourier attention proposed by FED-formerZhou et al. (2022) to capture frequency domain dependencies as Fourier transform and Wavelet transform has been widely applied in this task. Meanwhile, we utilize auto-correlation attention Wu et al. (2021) to capture time domain dependencies. Meanwhile, unlike the common multivariate time series prediction, in this task, the input signal and target signal are in different domains. Therefore, we need to capture the inner channel dependencies between the input and the target. We utilize the Two-Stage attention Zhang & Yan (2022) and replace the vanilla multi-head self-attention (MSA) Vaswani et al. (2017) with wavelet attention to explore the channel-wise time-frequency dependencies. To further improve model efficiencies and robustness, we applied the multitask learning framework to separate model predictions of valence and arousal with a shared backbone.

The major contribution of this paper are :

1. We propose a new State-of-the-art (SOTA) transformer-based solution for physiological signal emotion recognition.

2. We propose a new model architecture and framework for domain-inconsistent multivariate time series prediction.

3. Our proposed method achieved SOTA results on the Continuously Annotated Signals of Emotion (CASE) dataset. Sharma et al. (2019)

## 2 RELATED WORK

### 2.1 AUTOFORMER

Autoformer utilizing the Auto-correlation mechanism to generate long-term time series prediction.Wu et al. (2021) It introduces an Auto-correlation Attention to replaces the Multihead Self-Attention(MSA) in the vanilla transformer. It also proposed deep decomposition architecture to replace the vanilla transformer Encoder-Decoder architecture and series decomposition block to decompose the time series sequence into trend-cyclical and seasonal components.

#### 2.1.1 AUTO-CORRELATION ATTENTION

Auto-correlation Attention is a time domain attention based on auto-correlation mechanism. Let $\boldsymbol{Q}, \boldsymbol{K}, \boldsymbol{V}$ denote the query, key, and value inputs of the attention. It first capture time delay similarity via auto-correlation calculation based a fix $\tau$ length time delay window between $\boldsymbol{Q}$ and $\boldsymbol{K}$. It then choose most possible $k$ window periods and normalize them with Softmax. The final output will aggregate the product of rolling $\boldsymbol{V}$ based on the same $\tau$ length window and the normalized $k$ auto-correlation result of $\boldsymbol{Q}$ and $\boldsymbol{K}$. The process can be formulized as:

$$
\begin{aligned}
\mathrm{R}_{\boldsymbol{X}\boldsymbol{X}}(\tau) &= \lim_{L \to \infty} \frac{1}{L} \sum_{t=1}^{L} \boldsymbol{X}_t \boldsymbol{X}_{t-\tau} \\
\tau_1, \ldots, \tau_k &= \operatorname*{argTopk}_{\tau \in \{1,\ldots,\mathrm{L}\}} (\mathrm{R}_{\boldsymbol{Q},\boldsymbol{K}}(\tau)) \\
\hat{\mathrm{R}}_{\boldsymbol{Q},\boldsymbol{K}}(\tau_1), \ldots, \hat{\mathrm{R}}_{\boldsymbol{Q},\boldsymbol{K}}(\tau_k) &= \mathrm{SoftMax}(\hat{\mathrm{R}}_{\boldsymbol{Q},\boldsymbol{K}}(\tau_1), \ldots, \hat{\mathrm{R}}_{\boldsymbol{Q},\boldsymbol{K}}(\tau_k)) \\
\text{Auto-Correlation}(\boldsymbol{Q}, \boldsymbol{K}, \boldsymbol{V}) &= \sum_{i=1}^{k} \mathrm{Roll}(\boldsymbol{V}, \tau_i) \hat{\mathrm{R}}_{\boldsymbol{Q},\boldsymbol{K}}(\tau_i)
\end{aligned}
\tag{1}
$$

where $\boldsymbol{Q}, \boldsymbol{K}, \boldsymbol{V}$ are the attention input, $\tau$ is the time delay window length, $\mathrm{Roll}(\mathrm{X}, \tau)$ is the operation shift the element in $\boldsymbol{X}$ by $\tau$ length window and the element shift beyond the first position will re-introduced at the last position. In this study, we utilize the Auto-correlation Attention to decompose the pure time domain representation from the time-frequency representation from the encoder.

#### 2.1.2 SERIES DECOMPOSITION

The series decomposition block decomposes the series sequence into trend-cyclical and seasonal components. The decompose process can be presented as:

$$
\begin{aligned}
\boldsymbol{T} &= \mathrm{AvgPool}(\mathrm{Padding}(\boldsymbol{X})) \\
\boldsymbol{S} &= \boldsymbol{X} - \boldsymbol{T}
\end{aligned}
\tag{2}
$$

where $\boldsymbol{X} \in \mathbb{R}^{L \times d}$ is raw time series, $\boldsymbol{T}, \boldsymbol{S} \in \mathbb{R}^{L \times d}$ and are the decomposed trend and seasonal components. The following of the paper will denote this block as $\boldsymbol{T}, \boldsymbol{S} = \mathrm{SeriesDecomp}(\boldsymbol{X})$.

### 2.2 FEDFORMER

In Zhou et al. (2022), the authors proposed a new transformer based for long-term time series prediction that utilize the frequency domain representation. It inherit the same model architecture from Autoformer and but replace the auto-correlation attention in time domain with Fourier and Wavelet

attention in frequency domain. It also further improve series decomposition block with multiple window length instead of fix window length.

### 2.2.1 FOURIER ATTENTION

FEDformer model propose a pure frequency domain attention based on Fourier transform with global property. Let $\mathcal{F}$ and $\mathcal{F}^{-1}$ denote Fourier transform and inverse Fourier transform. The Fourier attention will first transform the attention input $\boldsymbol{Q}, \boldsymbol{K}, \boldsymbol{V}$ into frequency domain and perform attention mechanism via Fourier transform and transform the attention representation back to time domain via inverse Fourier transform. The process of the Fourier attention could be shown as:

$$\text{Fourier-Attention}(\boldsymbol{Q}, \boldsymbol{K}, \boldsymbol{V}) = \mathcal{F}^{-1}(\text{Padding}(\sigma(\mathcal{F}(\boldsymbol{Q}) \cdot \mathcal{F}(\boldsymbol{K})^T) \cdot \mathcal{F}(\boldsymbol{V}))) \quad (3)$$

where $\mathcal{F}$ denotes random selecting modes Fourier transform, and $\sigma$ denotes the activation function. Details of selecting method could be found in Zhou et al. (2022).

In this study, the Fourier Attention is implemented for decompose the pure frequency domain representation from time-frequency domain representation from the encoder.

### 2.2.2 WAVELET ATTENTION

Wavelet attention is another version of frequency enhanced attention proposed by FEDformer. It is a time and frequency attention with local property. The wavelet attention will first decompose the attention input $\boldsymbol{Q}, \boldsymbol{K}, \boldsymbol{V}$ in multiscale and perform attention mechanism on each scale and then reconstruct the attention representation. Let $\mathcal{F}$ and $\mathcal{F}^{-1}$ denote Wavelet decomposition and Wavelet reconstruction process. The Wavelet attention can be formulized as:

$$\text{Wavelet-Attention}(\boldsymbol{Q}, \boldsymbol{K}, \boldsymbol{V}) = \mathcal{W}^{-1}(\sigma(\mathcal{W}(\boldsymbol{Q}) \cdot \mathcal{W}(\boldsymbol{K}^T)) \cdot \mathcal{W}(\boldsymbol{V})) \quad (4)$$

where $\sigma$ denotes the activation function. Details of the multiwavelet implementation could be found in Zhou et al. (2022).

In this study, we utilize the wavelet attention to extract time-frequency representation from the encoder input and then decomposed by the time and frequency decoder into pure time domain and frequency domain representation.

### 2.3 MIXTURE OF EXPERTS SERIES DECOMPOSITION(MOE-DECOMPOSE)

FEDformer proposed a multi-kernel version of series decomposition block. The trend component $\boldsymbol{T}$ will be extracted from a set of $\text{AvgPool}$ layers and sum the result with weight. The redesigned trend can be presented as:

$$\boldsymbol{T} = \text{SoftMax}(\text{L}(\boldsymbol{X})) \times \text{P}(\boldsymbol{X}) \quad (5)$$

where $\text{SoftMax}(\text{L}(\boldsymbol{X}))$ denote the weights, and $\text{P}(\cdot)$ denote the set of $\text{AvgPool}$ layers. This block will be denoted as $\boldsymbol{T}, \boldsymbol{S} = \text{MOEDecomp}((\boldsymbol{X}))$ in rest of the paper.

### 2.4 CROSSFORMER

While most time series prediction model focuses on cross-time dependencies, recently,Zhang & Yan (2022) proposed a new embedding layer and attention framework which can capture both cross-time dependencies and cross-dimensional dependencies with patch segmentation.

### 2.4.1 DIMENSION WISE SEGMENT (DSW) EMBEDDING

The dimension wise segment (DSW) embedding will first segment the model input into fix length patches. To make embedding contains cross-dimension dependencies, DSW embedding embeds the input over patches axis, instead of feature(or channel) axis of the inputs. Let $\text{L}_{\text{seg}}$ denote the patch size (or segment length). The embedding could be shown as:

$$\boldsymbol{X}_{embed} = \text{E}_{segment}(\boldsymbol{X}_{inp}) + \text{E}_{pos} \quad (6)$$

where $\boldsymbol{X}_{inp} \in \mathbb{R}^{T \times C}$ denote the model input, and $T$ is sequence length and $C$ is the number of channels (or features). $\boldsymbol{X}_{embed} \in \mathbb{R}^{C \times \frac{T}{L_{seg}} \times D_{model}}$ denote embedded input and E are the learnable representations, where $D_{model}$ is the number of hidden units of the E. In this study, we customized DSW embedding and details about our implementation will be discussed in section 3.1.3

### 2.4.2 Two-Stage Attention (TSA)

To capture the both cross-time and cross-dimension dependencies from DSW embedded input, Crossformer proposed a new attention layer called Two-Stage Attention (TSA), which process the 2D embedded input through two different stages, cross-time stage and cross-dimension. In the cross-time stage, the attention is calculated along the time segment axis and in the cross-dimension stage the attention is calculated over the channel axis. Two-Stage Attention can be formulized as:

$$\boldsymbol{X}^{time} = \text{CTA}(\boldsymbol{X})$$
$$\boldsymbol{X}^{dim} = \text{CDA}(\boldsymbol{X}^{time}) \tag{7}$$

where $\boldsymbol{X}^{dim}$ denotes the output the TSA block.

Details about the vanilla implementation of TSA are described in Appendix A.2. Our customized TSA implementation of Frequency enhanced TSA, Frequency Domain TSA and Time Domain TSA are described in Appendix A.3,A.4, and A.5 repetively.

## 3 Methodology

We first convert the physiological signal emotion recognition task to a multivariate time series sequence-to-sequence prediction task. Denote the input physiological signal as $\boldsymbol{I}$, the target valence and arousal signal as $\boldsymbol{V}, \boldsymbol{A}$, and the sample rate difference between them as $\Delta = $ (Input sample rate)/(Output sample rate). The common long term time series prediction task will have encoder input with $i$ time step and decoder input with $i/2 + o$ time step, where $i$ is defined as the sequence length, $i/2$ is defined as the label length, and $o$ is defined as the prediction length. Due to the inconsistency between the input and target domains in this task, we change the label length to $o/2$ instead. With the sample rate difference $\Delta$, the sequence length $i$ will be $\Delta \times (o/2 + o)$.

### 3.1 Data Preprocessing

The data prepossessing of this model includes 3 parts: signal preprocessing, patch segmentation, and sequence embedding.

### 3.1.1 Signal Preprocessing

We utilize the same preprocessing implementation of D'Amelio et al. (2023). Including 6 filtered physiological signals (3 ECG signals, EMG , GSR), 2 raw signals (BVP, SKT) and 5 additional continuous signals extracted from physiological signals: 3 from EDA (phasic component, sparse SMNA driver of the phasic component, and tonic component), 1 from ECG (NN intervals) and 1 from respiration (instantaneous respiratory rate). All the filtered signals are processed following the recommendation from Makowski et al. (2021). The details of the 5 additional continuous signals can be found in D'Amelio et al. (2023).

### 3.1.2 Patch Segmentation and Input projection

In this study, the sample rate difference between the input (1kHz) and the target (20Hz) signals is 50. As a result, we used a fixed length 50 to segment the input signal to match the target signal dimension. Meanwhile the moving average kernel of Series Decomposition Block for the input signal is also fixed as 50.

Due to the input signal and the target signal having different channels, we use two linear blocks to project the decomposed input signals seasonal and trend to the same channel dimension with the target. The seasonal will be first project to target channel dimension then segment to patches with size as 50 to keep most original channel-wise distribution. The trend will first segment to patches with size as 50 and then project to the target channel dimension to keep most original timestep-wise

distribution. The process can be formulized as follows:

$$\boldsymbol{S}_{inp}, \boldsymbol{T}_{inp} = \text{MOEDecomp}(\boldsymbol{X}_{inp})$$
$$\boldsymbol{S}_{proj} = \text{LinearBlock}(\boldsymbol{S}_{inp})$$
$$\boldsymbol{T}_{seg} = \sum_{k=0}^{\frac{i}{50}} \boldsymbol{T}_{inp}^{(k,...,k+50)} \tag{8}$$
$$\boldsymbol{T}_{proj} = \text{LinearBlock}(\boldsymbol{T}_{seg})$$

where $\boldsymbol{S}_{inp}, \boldsymbol{T}_{inp} \in \mathbb{R}^{i \times C_{inp}}$ are the seasonal and trend decompose from the input signals $X_{inp}$ (the preprocessed physiological signals), $i$ is the input (sequence) length and $C_{enc}$ denote the channel dimension of the encoder input. $\boldsymbol{T}_{seg} \in \mathbb{R}^{o \times C_{enc}}$, is the segmented trend and $o$ is the output(prediction) length $\boldsymbol{S}_{proj}, \boldsymbol{T}_{proj} \in \mathbb{R}^{o \times C_{dec}}$ are the projected seasonal and trend and $C_{enc}$ denote the channel dimension of the decoder input.

### 3.1.3 SEQUENCE EMBEDDING

We combine DSW embedding with the time feature embedding generated from the timestamp of the signals. We drop the positional embedding including in the Crossformer implementation. The reasons are: 1) The sequential information is inherited in the time feature embedding; 2) From the FEDformer implementation and our own practical and empirical results, adding positional embedding with time feature embedding will cause unnecessary overfitting which eventually decreased the model performance.

For the patch size (or segment length) of DSW embedding, we use 1,2,5,25,50 for encoder input and 1,3,5,15 for decoder input as different initial time domain resolution. Then all different patch size embedding are sum together to generate a dynamic resolution DSW embedding. The embedding process is shown as following:

$$\boldsymbol{X}_{enc} = \text{E}_{DSW}(\boldsymbol{X}_{inp}) + \text{E}_{time}(timestamp_{enc})$$
$$\boldsymbol{S}_{dec} = \sum_{k=0}^{\frac{i}{50}} \boldsymbol{S}_{proj}^{(k,...,k+50)} \tag{9}$$
$$\boldsymbol{X}_{dec} = \text{E}_{DSW}(\boldsymbol{S}_{dec}) + \text{E}_{time}(timestamp_{dec})$$

where E denotes the learnable embedding layer, $\boldsymbol{X}_{enc} \in \mathbb{R}^{i \times D_{model}}$ denotes the encoder input, and $\boldsymbol{S}_{proj} \in \mathbb{R}^{i \times C_{dec}}$ and $\boldsymbol{S}_{dec} \in \mathbb{R}^{o \times C_{dec}}$ denote projected seasonal part and segmented seasonal. $\boldsymbol{X}_{dec} \in \mathbb{R}^{o \times D_{model}}$ denotes the input of the decoder.

### 3.2 MODEL ARCHITECTURE

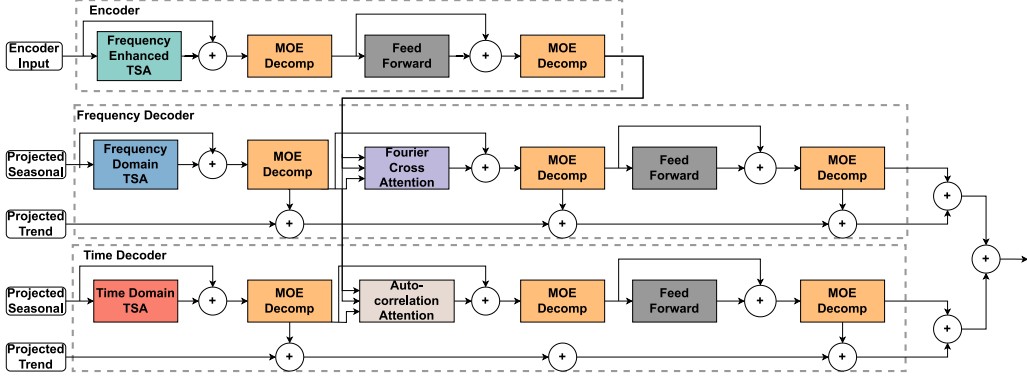

Figure 1: Overall Model Architecture

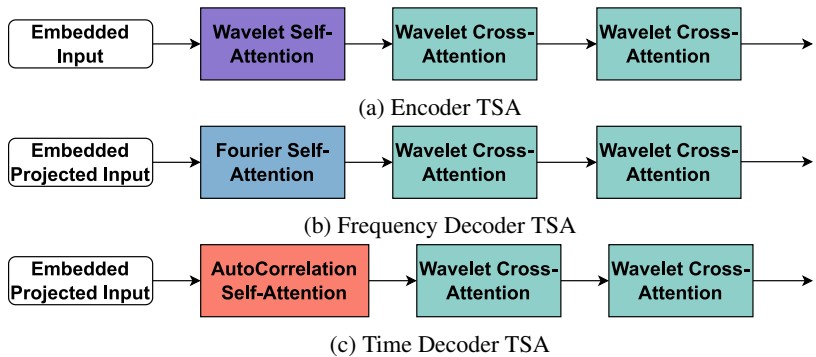

Figure 2: TSA architecture for encoder and decoder

The proposed model architecture is shown in Figure1. It includes one encoder block and two decoder blocks. The general architecture is similar with FEDformer. However, instead of plain frequency domain (Fourier and Wavelet) and time domain(Auto-correlation) attention, we reconstructed the self-attention block with TSA framework. The structures of the reconstructed self-attention block of the encoder and the decoders are shown in Figure2. Details of the encoder block and decoder blocks will be introduced in Section 3.2.1 and Section 3.2.2. In section 3.2.3, we introduce the technique to handle the domain inconsistency in this task. The loss function and the multitask framework will be introduced in Sections 3.3 and 3.4.

### 3.2.1 ENCODER

The encoder contains 3 major parts, Frequency Enhanced TSA, MOE-Decompose, and FeedForward layer, which can be formulized as:

$$
\begin{aligned}
\boldsymbol{S}_{enc}^{l,1}, {}_- &= \text{MOEDecomp}(\text{FrequencyEnhanceTSA}(X_{enc}^{l-1}) + \boldsymbol{X}_{enc}^{l-1}) \\
\boldsymbol{S}_{enc}^{l,2}, {}_- &= \text{MOEDecomp}(\text{FeedForward}(\boldsymbol{S}_{enc}^{l,1}) + \boldsymbol{S}_{enc}^{l,1}) \\
\boldsymbol{X}_{enc}^{l} &= \boldsymbol{S}_{enc}^{l,2}
\end{aligned}
\tag{10}
$$

where $\boldsymbol{S}_{enc}^{l,k} k \in 1, 2$ denote the seasonal part from the k-th decomposition block, and $l$ denotes the l-th encoder layer of the multi-layer structure. $_-$ denotes the dropped trend component. The frequency enhanced TSA has the similar structure with the vanilla TSA(detail structure in section 2.4.2) but replacing the MSA in Cross-time stage with Wavelet Self-Attention and the MSAs in the Cross-Dimension stage with Wavelet Cross-Attention as shown in Figure 2a and details implementation are described in Appendix A.3. The output of the encoder will be the seasonal component of the second decomposition block. By replacing the MSA with wavelet attention, the encoder will generate local time-frequency domain representation from both timestep-wise and channel-wise.

### 3.2.2 DECODER

There are two decoders in the structure to decomposed the time-frequency domain representation from encoder into pure frequency domain and time domain representations. The decoding process can be formulized as:

$$
\begin{aligned}
\boldsymbol{X}_{dec}^{l(freq)}, \boldsymbol{T}_{dec}^{l(freq)} &= \text{FrequencyDecoder}(\boldsymbol{X}_{dec}) \\
\boldsymbol{X}_{dec}^{l(time)}, \boldsymbol{T}_{dec}^{l(time)} &= \text{TimeDecoder}(\boldsymbol{X}_{dec})
\end{aligned}
\tag{11}
$$

where Frequency Decoder can be presented as:

$$\boldsymbol{S}_{dec}^{l,1(freq)}, \boldsymbol{T}_{dec}^{l,1(freq)} = \text{MOEDecomp}(\text{FrequencyDomainTSA}(\boldsymbol{X}_{dec}^{l-1}) + \boldsymbol{X}_{dec}^{l-1})$$
$$\boldsymbol{S}_{dec}^{l,2(freq)}, \boldsymbol{T}_{dec}^{l,2(freq)} = \text{MOEDecomp}(\text{FourierCrossAttention}(\boldsymbol{S}_{dec}^{l,1(freq)}, \boldsymbol{X}_{enc}^{N}) + \boldsymbol{S}_{dec}^{l,1(freq)})$$
$$\boldsymbol{S}_{dec}^{l,3(freq)}, \boldsymbol{T}_{dec}^{l,3(freq)} = \text{MOEDecomp}(\text{FeedForward}(\boldsymbol{S}_{dec}^{l,2(freq)}) + \boldsymbol{S}_{dec}^{l,2(freq)}) \quad (12)$$
$$\boldsymbol{X}_{dec}^{l(freq)} = \boldsymbol{S}_{dec}^{l,3(freq)}$$
$$\boldsymbol{T}_{dec}^{l(freq)} = \boldsymbol{T}_{dec}^{l-1(freq)} + \boldsymbol{W}_{l,1} \times \boldsymbol{T}_{dec}^{l,1(freq)} + \boldsymbol{W}_{l,2} \times \boldsymbol{T}_{dec}^{l,2(freq)} + \boldsymbol{W}_{l,3} \times \boldsymbol{T}_{dec}^{l,3(freq)}$$

The Time Decoder will be similar with Frequency Decoder but replacing the Frequency Domain Decoder to Time Domain Decoder, and Fourier Cross Attention to Auto-correlation attention.

The structures of Frequency and Time domain TSA are shown in Figure 2b and 2c respectively and details implementation are described in Appendix A.4 and A.5 repectively. Frequency domain TSA replaces the Cross-Time Attention with Fourier Attention, and time domain TSA replaces it with Auto-correlation attention. Both frequency and time domain TSA replace MSAs in the Cross-Dimension attention with wavelet cross-attention.

The decoders will generate pure frequency and time domain representations along each timestep and time-frequency domain representations over each channel. Then, generate pure frequency and time domain representation with encoder output through the Fourier Cross Attention and Auto-correlation attention, respectively. The final decoder output will be sum of both frequency decoder and time decoder after multiplied the domain adaptive ratio, which will be introduced in the following section.

### 3.2.3 DOMAIN ADAPTATION RATIO

Unlike common multivariate time series prediction, in this task, the feature signal and the target signal have different domains. To address this, we implemented a learnable adaptive ratio. We first project the seasonality and trend extracted from the input physiological signals to the same channel dimension with the target signal via 2 linear blocks(each for valence and arousal respectively). Then, we calculate the difference ratio between the feature and target signal over the label length and the difference ratio between the feature signal over the label length and the prediction length. The final domain adaptation ratio will be the product of these two ratios. The formula of the domain adaptive ratio is presented in Appendix A.1 and the final model outputs will be:

$$\boldsymbol{X}_{dec}^{(freq)} = \text{ratio}_{\boldsymbol{S}} \times \boldsymbol{X}_{dec}^{l(freq)} + \text{ratio}_{\boldsymbol{T}} \times \boldsymbol{T}_{dec}^{l(freq)}$$
$$\boldsymbol{X}_{dec}^{(time)} = \text{ratio}_{\boldsymbol{S}} \times \boldsymbol{X}_{dec}^{l(time)} + \text{ratio}_{\boldsymbol{T}} \times \boldsymbol{T}_{dec}^{l(time)} \quad (13)$$
$$\boldsymbol{X}_{dec} = \frac{1}{2} \times \boldsymbol{X}_{dec}^{(freq)} + \frac{1}{2} \times \boldsymbol{X}_{dec}^{(time)}$$

where $\text{ratio}_{\boldsymbol{S}}, \text{ratio}_{\boldsymbol{T}}$ denote the seasonal and the trend adaptive ratio respectively, and $\boldsymbol{X}_{dec}$ denote the model output.

### 3.3 LOSS FUNCTION

Dilate loss is selected as loss function for this study.Le Guen & Thome (2019) It can address the shape and temporal distortion of time series prediction. It contains two parts $Loss_{shape}$ and $Loss_{temproal}$, which can be formulized as:

$$L_{Dilate} = \alpha L_{shape} + (1-\alpha)L_{temproal} \quad (14)$$

The shape loss $L_{shape}$ is based on the Dynamic Time Warping (DTW), and the temproal loss is calualting Time Distortion Index (TDI) for temporal misalignment estimation. Details of the formulas can be found in Le Guen & Thome (2019).

### 3.4 MULTI-TASK LEARNING

We implement the multitask framework to separately predict valence and arousal. The top layer of the decoders will generate separate $\boldsymbol{X}_{dec}^{l(valence)}, \boldsymbol{T}_{dec}^{l(valence)}$ for valence and $\boldsymbol{X}_{dec}^{l(arousal)}, \boldsymbol{T}_{dec}^{l(arousal)}$

for arousal. The final model output $\boldsymbol{X}_{dec}$ will be:

$$
\begin{aligned}
\boldsymbol{X}_{dec}^{l} &= \text{Concat}(\boldsymbol{X}_{dec}^{l(valence)}, \boldsymbol{X}_{dec}^{l(arousal)}) \\
\boldsymbol{T}_{dec}^{l} &= \text{Concat}(\boldsymbol{T}^{l(valence)}, \boldsymbol{T}_{dec}^{l(arousal)}) \\
\boldsymbol{X}_{dec} &= \boldsymbol{X}_{dec}^{l} + \boldsymbol{T}_{dec}^{l}
\end{aligned}
\tag{15}
$$

The loss function of valence and arousal predictions will be calculated separately. To combine the loss of valence $L_{valence}$ and the loss of arousal $L_{arousal}$, we implement the joint loss $L_{joint}$ proposed in Kendall et al. (2018). The formula is:

$$
L_{joint} = \frac{1}{\sigma_{valence}^2} L_{valence} + \frac{1}{\sigma_{arousal}^2} L_{arousal} + \log \sigma_{valence}^2 + \log \sigma_{arousal}^2
\tag{16}
$$

where $\sigma$ is noise parameter.

## 4 EXPERIMENTS & RESULTS

### 4.1 DATASET

The evaluation dataset used in this study is the Continuously Annotated Signals of Emotion (CASE) dataset.Sharma et al. (2019) It contains the physiological and annotation data from 30 participants, including 15 male and 15 female. The continuous valence and arousal annotations are reported by the participant via a joystick simultaneously while watching various videos. The physiological data is also collected while the participants are watching videos from electrocardiograph (ECG), blood volume pulse(BVP), electromyography(EMG) (3 channels), Galvanic Skin Response(GSR) (or Electrodermal(EDA)), respiration(RSP) and skin temperature(SKT) sensors, in total 8 raw physiological measurements. The sample rate of the physiological signals is 1000Hz and sample rate of the annotation is 20Hz.

### 4.2 EXPERIMENT SETTINGS

We follows the experiment settings of the Emotion Physiology and Experience Collaboration (EPiC) challenge. The CASE dataset will be split into four different test scenarios. In total, there are 240 files which are generated from 30 participants watch eight videos, categorized in four emotion, "scary", "relaxing", "boring", and "amusing". Annotation data is range from 0.5-9.5. Details of each scenario setting will be introduced in the following section.

- **Across-time scenario:** In this scenario, the 240 data files are divided into training data and testing data based on the timestamp. The former part of the data files are selected as the training data and the latter part of the data file become the testing data. This scenario is aiming to examine the model's ability of capturing the time dependencies between the samples.

- **Across-subject scenario:** In across-subject scenario, 30 participants are divided into 5 folds, where each folds contains 24 participants as the training data (192 files in total) and 6 participants as the testing data (48 files). This scenario aims to evaluate the model's subject independence performance.

- **Across-elicitor scenario:** In this scenario, the eight videos which are labeled in four different emotions will be divided into four folds. Each fold contains six videos categorized in three emotions (180 files) as training data and two videos categorized in the left emotion (60 files) as testing data. This scenario aims to test the model's generalization ability towards unseen emotion.

- **Across-version scenario:** In across-version scenario, eight videos categorized in four emotions are separated into 2 folds. Each fold contain four videos that categorized in all four emotions (120 files) as training data and the other four videos categorized in all four emotions (120 files) are treated as different version to be testing data. This scenario aims to verify if the model can generalize the emotional state learned from one source to another source elicits the same emotional state.

Table 1: RMSE of All scenarios. Comparing with results from AutoGluon(Dollack et al., 2023) and RF or XGBoost (D'Amelio et al., 2023), Multi-scale Transformer(Vu et al., 2023)

| Scenarios | Fold | Ours | | AutoGluon | | RF or XGBoost | | Multi-scale Transformer | |
|---|---|---|---|---|---|---|---|---|---|
| | | Valence | Arousal | Valence | Arousal | Valence | Arousal | Valence | Arousal |
| Across-time scenario | | **0.84** | **0.69** | 0.95 | 0.91 | 0.87 | 0.85 | 1.50 | 1.64 |
| Across-subject scenario | 0 | 0.62 | 0.59 | 1.14 | 1.14 | - | - | | |
| | 1 | 0.49 | 0.53 | 1.17 | 1.03 | - | - | - | - |
| | 2 | 0.53 | 0.64 | 1.21 | 1.18 | - | - | - | - |
| | 3 | 0.48 | 0.48 | 1.26 | 0.92 | - | - | - | - |
| | 4 | 0.45 | 0.50 | 1.13 | 0.74 | - | - | - | - |
| Scenario level | | **0.51** | **0.55** | 1.18 | 1.00 | 1.36 | 1.17 | 1.34 | 1.35 |
| Across-elicitor scenario | 0 | 1.93 | 2.40 | 2.36 | 2.29 | - | - | | |
| | 1 | 0.94 | 0.65 | 1.27 | 0.92 | - | - | - | - |
| | 2 | 0.73 | 0.86 | 0.92 | 1.35 | - | - | - | - |
| | 3 | 0.89 | 0.62 | 1.15 | 1.20 | - | - | - | - |
| Scenario level | | **1.12** | **1.13** | 1.42 | 1.44 | 1.49 | 1.63 | 1.51 | 1.51 |
| Across-version scenario | 0 | 0.84 | 0.81 | 1.06 | 1.00 | - | - | - | - |
| | 1 | 0.96 | 1.04 | 1.54 | 1.60 | - | - | - | - |
| Scenario level | | **0.90** | **0.93** | 1.30 | 1.30 | 1.36 | 1.52 | 1.37 | 1.35 |

## 4.3 MODEL SETTINGS

The sequence length for all scenarios is 2250, the label length is 15 and the prediction length is 30. The batch size for all scenarios set as 16. We use the AdamW optimizer with $1e^{-5}$ weight decay. Models for each scenario are trained 15 epochs.

## 4.4 RESULTS

The inference process of our model require pseudo label. In Across-time scenario, the pseudo label will generated from the label length segment of the end of the training data with same subject id an video id. In Across-subject scenario, we first explore the training data has the same video ID as the testing data, and then the gender of training subject and with the testing subject. In Across-elicitor scenario, our optimal result achieved from utilizing the training data not containing 'scary' elicitor. The poor performance of fold 0 RMSE in Across-elicitor scenario due to the testing data are the videos labeled as 'scary'. In Across-version scenario, our optimal result doesn't achieve from using the pseudo label from exact matched version. In some cases, the pseudo label from other elicitor yield better results.

Table1 presents the results of four scenarios. AutoGluon solution proposed by Dollack et al. (2023) achieves the first place in EPiC challenge, RF/XGBoost solution proposed by D'Amelio et al. (2023) achieves third place in the challenge , and the multi-scale Transformer-based solution proposed by Vu et al. (2023). Our results outperforms the other three solutions in all four scenarios, which proves our solution is very promising and can achieve SOTA result in all the test scenarios.

## 5 CONCLUSION

In this study we proposed a SOTA transformer based solution for EPiC challenge. Our proposed model achieve SOTA result in all four scenarios. It proves our proposed method is very promising. For future work, we will improve the model architecture to make it more adaptive to the domain inconsistent situation.

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

# A APPENDIX

## A.1 FORMULA OF DOMAIN ADAPTIVE RATIO

The domain adaptive ratio can be formulized as :

$$
\begin{aligned}
\boldsymbol{S}_{target}, \boldsymbol{T}_{target} &= \text{MOEDecomp}(\boldsymbol{X}_{target}^{(0,\dots,\frac{o}{2})}) \\
\text{ratio}_S^{(dec)} &= \frac{\frac{1}{n}\sum S_{target}}{\frac{1}{n}\sum S_{dec}^{(0,\dots,\frac{o}{2})}} \\
\text{ratio}_S^{(enc)} &= \frac{\frac{1}{n}\sum S_{dec}^{(0,\dots,\frac{o}{2})}}{S_{dec}^{(\frac{o}{2},\dots,o)}} \\
\text{ratio}_S &= \text{ratio}_S^{(enc)} \times \text{ratio}_S^{(dec)} \\
\text{ratio}_T^{(dec)} &= \frac{\frac{1}{n}\sum T_{target}}{\frac{1}{n}\sum T_{dec}^{(0,\dots,\frac{o}{2})}} \\
\text{ratio}_T &= \text{ratio}_T^{(dec)}
\end{aligned}
\tag{17}
$$

where the $ratio_S$ $ratio_T$ denote the adaptive ratio for seasonal and trend respectively. $X_{raw}^{(0,\dots,\frac{o}{2})(target)}$ denote the target signal with the label length.

## A.2 VANILLA TSA IMPLEMENTATION

The Cross Time stage Attention(CTA) is structured as:

$$
\begin{aligned}
\hat{\boldsymbol{X}}_t &= \text{LayerNorm}(\boldsymbol{X}_t + \text{MSA}(\boldsymbol{X}_t, \boldsymbol{X}_t, \boldsymbol{X}_t)) \\
\boldsymbol{X}_t^{time} &= \text{LayerNorm}(\hat{\boldsymbol{X}}_t + \text{LinearBlock}(\hat{\boldsymbol{X}}_t))
\end{aligned}
\tag{18}
$$

where $\boldsymbol{X}_t, \boldsymbol{X}_t^{time} \in \mathbb{R}^{\frac{T}{L_{seg}} \times D_{model}}$ denote for the input and output of the cross-time stage attention, LayerNorm and MSA denote the layer normalization layer and vanilla multi-head self attention.

and Cross Dimension stage Attention(CDA) is presented as:

$$
\begin{aligned}
\boldsymbol{B} &= \text{MSA}_1^{dim}(\boldsymbol{R}, \boldsymbol{X}_d^{time}, \mathbf{X}_d^{time})) \\
\bar{\boldsymbol{X}}^{dim} &= \text{MSA}_2^{dim}(\boldsymbol{X}_d^{time}, \boldsymbol{B}, \boldsymbol{B})) \\
\hat{\boldsymbol{X}}^{dim} &= \text{LayerNorm}(\boldsymbol{X}_d^{time} + \bar{\boldsymbol{X}}^{dim}) \\
\boldsymbol{X}^{dim} &= \text{LayerNorm}(\hat{\boldsymbol{X}}^{dim} + \text{LinearBlock}(\hat{\boldsymbol{X}}^{dim}))
\end{aligned}
\tag{19}
$$

where $\boldsymbol{R} \in \mathbb{R}^{f \times D_{model}}$ is learnable vector which server as router to reduce the computation complexity and $f$ is a constant number denote the number of the routers. $\boldsymbol{X}_d^{time} \in \mathbb{R}^{C \times D_{model}}$ denotes the reshaped cross-time stage attention output as the cross-dimension stage attention input, and $\boldsymbol{X}^{dim} \in \mathbb{R}^{C \times \frac{T}{L_{seg}} \times D_{model}}$ denotes the cross dimension stage output. The $\boldsymbol{X}^{dim}$ will also be the final output of the complete TSA block.

## A.3 FREQUENCY ENHANCED TSA IMPLEMENTATION

The Cross Time stage Attention(CTA) is structured as:

$$\hat{\boldsymbol{X}}_t = \text{LayerNorm}(\boldsymbol{X}_t + \text{WaveletSelfAttention}(\boldsymbol{X}_t, \boldsymbol{X}_t, \boldsymbol{X}_t))$$
$$\boldsymbol{X}_t^{time} = \text{LayerNorm}(\hat{\boldsymbol{X}}_t + \text{LinearBlock}(\hat{\boldsymbol{X}}_t)) \tag{20}$$

where $\boldsymbol{X}_t, \boldsymbol{X}_t^{time} \in \mathbb{R}^{\frac{T}{L_{seg}} \times D_{model}}$ denote for the input and output of the cross-time stage attention, LayerNorm and MSA denote the layer normalization layer and vanilla multi-head self attention.

and Cross Dimension stage Attention(CDA) is presented as:

$$\boldsymbol{B} = \text{WaveletCrossAttention}_1^{dim}(\boldsymbol{R}, \boldsymbol{X}_d^{time}, \boldsymbol{\mathsf{X}}_d^{time}))$$
$$\bar{\boldsymbol{X}}^{dim} = \text{WaveletCrossAttention}_2^{dim}(\boldsymbol{X}_d^{time}, \boldsymbol{B}, \boldsymbol{B}))$$
$$\hat{\boldsymbol{X}}^{dim} = \text{LayerNorm}(\boldsymbol{X}_d^{time} + \bar{\boldsymbol{X}}^{dim}) \tag{21}$$
$$\boldsymbol{X}^{dim} = \text{LayerNorm}(\hat{\boldsymbol{X}}^{dim} + \text{LinearBlock}(\hat{\boldsymbol{X}}^{dim}))$$

where $\boldsymbol{R} \in \mathbb{R}^{f \times D_{model}}$ is learnable vector which server as router to reduce the computation complexity and $f$ is a constant number denote the number of the routers. $\boldsymbol{X}_d^{time} \in \mathbb{R}^{C \times D_{model}}$ denotes the reshaped cross-time stage attention output as the cross-dimension stage attention input, and $\boldsymbol{X}^{dim} \in \mathbb{R}^{C \times \frac{T}{L_{seg}} \times D_{model}}$ denotes the cross dimension stage output. The $\boldsymbol{X}^{dim}$ will also be the final output of the complete TSA block.

## A.4 FREQUENCY DOMAIN TSA IMPLEMENTATION

The Cross Time stage Attention(CTA) is structured as:

$$\hat{\boldsymbol{X}}_t = \text{LayerNorm}(\boldsymbol{X}_t + \text{FourierSelfAttention}(\boldsymbol{X}_t, \boldsymbol{X}_t, \boldsymbol{X}_t))$$
$$\boldsymbol{X}_t^{time} = \text{LayerNorm}(\hat{\boldsymbol{X}}_t + \text{LinearBlock}(\hat{\boldsymbol{X}}_t)) \tag{22}$$

where $\boldsymbol{X}_t, \boldsymbol{X}_t^{time} \in \mathbb{R}^{\frac{T}{L_{seg}} \times D_{model}}$ denote for the input and output of the cross-time stage attention, LayerNorm and MSA denote the layer normalization layer and vanilla multi-head self attention.

and Cross Dimension stage Attention(CDA) is presented as:

$$\boldsymbol{B} = \text{WaveletCrossAttention}_1^{dim}(\boldsymbol{R}, \boldsymbol{X}_d^{time}, \boldsymbol{\mathsf{X}}_d^{time}))$$
$$\bar{\boldsymbol{X}}^{dim} = \text{WaveletCrossAttention}_2^{dim}(\boldsymbol{X}_d^{time}, \boldsymbol{B}, \boldsymbol{B}))$$
$$\hat{\boldsymbol{X}}^{dim} = \text{LayerNorm}(\boldsymbol{X}_d^{time} + \bar{\boldsymbol{X}}^{dim}) \tag{23}$$
$$\boldsymbol{X}^{dim} = \text{LayerNorm}(\hat{\boldsymbol{X}}^{dim} + \text{LinearBlock}(\hat{\boldsymbol{X}}^{dim}))$$

where $\boldsymbol{R} \in \mathbb{R}^{f \times D_{model}}$ is learnable vector which server as router to reduce the computation complexity and $f$ is a constant number denote the number of the routers. $\boldsymbol{X}_d^{time} \in \mathbb{R}^{C \times D_{model}}$ denotes the reshaped cross-time stage attention output as the cross-dimension stage attention input, and $\boldsymbol{X}^{dim} \in \mathbb{R}^{C \times \frac{T}{L_{seg}} \times D_{model}}$ denotes the cross dimension stage output. The $\boldsymbol{X}^{dim}$ will also be the final output of the complete TSA block.

## A.5 TIME DOMAIN TSA IMPLEMENTATION

The Cross Time stage Attention(CTA) is structured as:

$$\hat{\boldsymbol{X}}_t = \text{LayerNorm}(\boldsymbol{X}_t + \text{Auto-CorrelationAttentio}(\boldsymbol{X}_t, \boldsymbol{X}_t, \boldsymbol{X}_t))$$
$$\boldsymbol{X}_t^{time} = \text{LayerNorm}(\hat{\boldsymbol{X}}_t + \text{LinearBlock}(\hat{\boldsymbol{X}}_t)) \tag{24}$$

where $\boldsymbol{X}_t, \boldsymbol{X}_t^{time} \in \mathbb{R}^{\frac{T}{L_{seg}} \times D_{model}}$ denote for the input and output of the cross-time stage attention, LayerNorm and MSA denote the layer normalization layer and vanilla multi-head self attention.

and Cross Dimension stage Attention(CDA) is presented as:

$$\boldsymbol{B} = \text{WaveletCrossAttention}_1^{dim}(\boldsymbol{R}, \boldsymbol{X}_d^{time}, \mathbf{X}_d^{time}))$$
$$\bar{\boldsymbol{X}}^{dim} = \text{WaveletCrossAttention}_2^{dim}(\boldsymbol{X}_d^{time}, \boldsymbol{B}, \boldsymbol{B}))$$
$$\hat{\boldsymbol{X}}^{dim} = \text{LayerNorm}(\boldsymbol{X}_d^{time} + \bar{\boldsymbol{X}}^{dim}) \qquad (25)$$
$$\boldsymbol{X}^{dim} = \text{LayerNorm}(\hat{\boldsymbol{X}}^{dim} + \text{LinearBlock}(\hat{\boldsymbol{X}}^{dim}))$$

where $\boldsymbol{R} \in \mathbb{R}^{f \times D_{model}}$ is learnable vector which server as router to reduce the computation complexity and $f$ is a constant number denote the number of the routers. $\boldsymbol{X}_d^{time} \in \mathbb{R}^{C \times D_{model}}$ denotes the reshaped cross-time stage attention output as the cross-dimension stage attention input, and $\boldsymbol{X}^{dim} \in \mathbb{R}^{C \times \frac{T}{L_{seg}} \times D_{model}}$ denotes the cross dimension stage output. The $\boldsymbol{X}^{dim}$ will also be the final output of the complete TSA block.

## A.6 PREDICTION VISUALIZATION

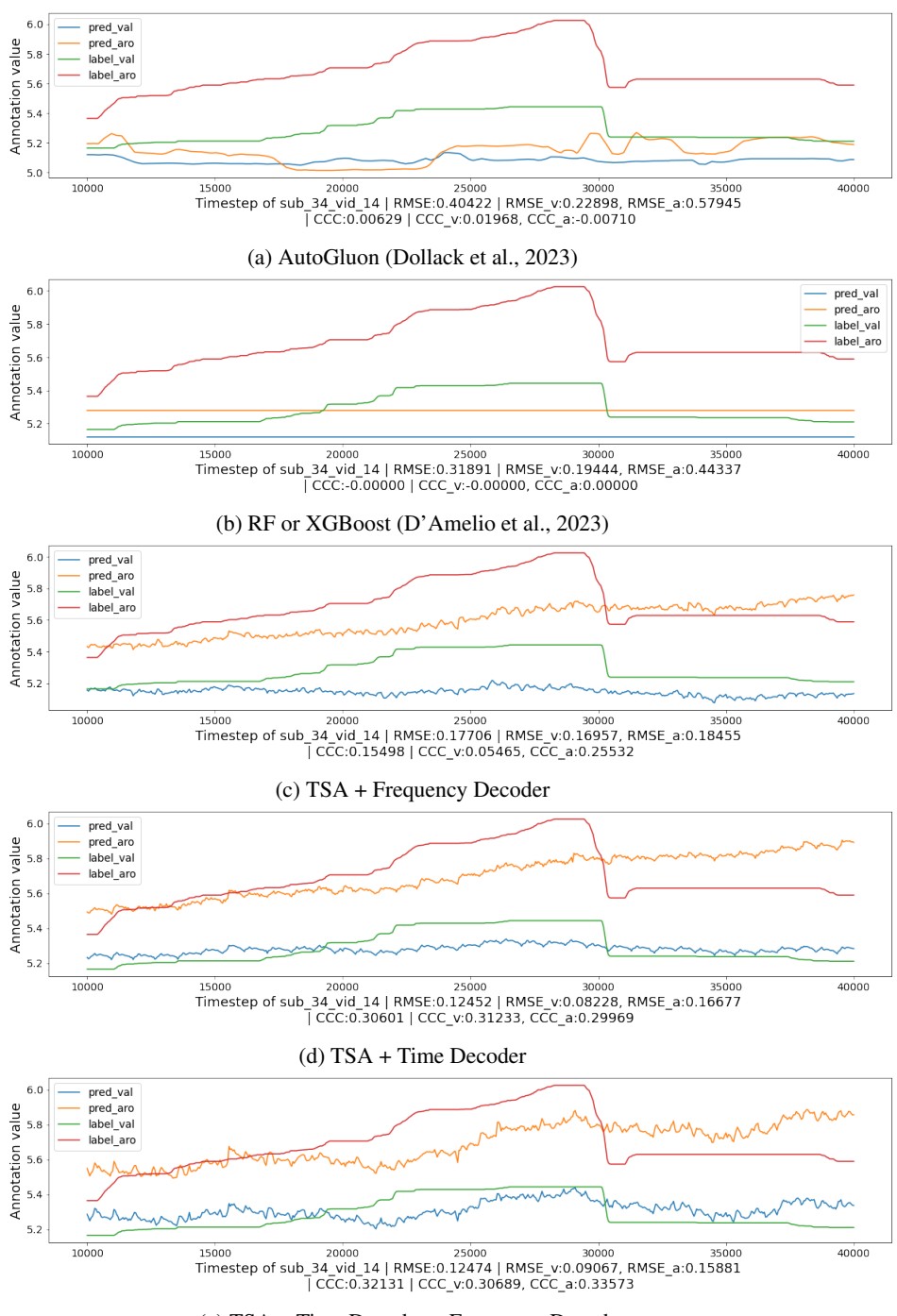

(a) AutoGluon (Dollack et al., 2023)

(b) RF or XGBoost (D'Amelio et al., 2023)

(c) TSA + Frequency Decoder

(d) TSA + Time Decoder

(e) TSA + Time Decoder + Frequency Decoder

Figure 3: Prediction visualization of different model. Two evaluation metrics of the prediction shows in the caption below the graph, root mean square error (RMSE) and concordance correlation coefficient (CCC)

