# OpenReview forum: "Decompose Time and Frequency Dependencies: Multivariate Time Series Physiological Signal Emotion Recognition"
_ICLR.cc/2024/Conference — Submitted to ICLR 2024_

### Official Review · Reviewer_zx9B · 2023-10-29

**Soundness:** 2 fair
**Presentation:** 2 fair
**Contribution:** 3 good
**Rating:** 5
**Confidence:** 4

**Summary:**

This paper introduces an end-to-end physiological signal emotion recognition model, and it achieved first place in all four tracks of the EPiC Challenge. Building upon the FED-Former, the model incorporates Two-Stage Attention (TSA) to capture temporal dependencies in the data. The paper implements multi-task learning for valence and arousal prediction tasks. The main contributions of the paper are as follows: 1. It proposes an end-to-end emotion recognition model based on physiological signals. 2. It enhances the FED-Former model to capture temporal dependencies between data. 3. The model performs exceptionally well across four tasks: Across-time scenario, Across-subject scenario, Across-elicitor scenario, and Across-version scenario.

**Strengths:**

#originality:
This paper demonstrates a good level of originality. Firstly, it addresses a problem in the field related to long-term time series forecasting in the time domain. Secondly, the proposed solution is simple yet effective. Although its overall structure is quite similar to FED-Former, the use of Two-stage Attention proves to be innovative and efficient. Furthermore, it performs well in scenarios across all four tasks, highlighting the model's originality.

#quality:
The quality of this work is relatively mediocre, mainly due to issues with experimental rigor, especially the lack of comprehensive model comparisons and tasks. Although the paper shines in terms of the model's performance, conducting more comparisons with traditional tasks would significantly enhance the paper's quality. The methodological rigor and reliability require improvement. There are also concerns regarding the overall presentation of results, figures, and tables, as there is a lack of graphical representation, and the experimental results are relatively limited.

#clarity:
The clarity of this paper is relatively good, but there is still room for improvement. The content is well-structured, with logical connections throughout. However, there are some drawbacks: 1. The representation of certain letters is not very intuitive and may lead to confusion. 2. There is a lack of illustrations, making the descriptions less visually informative. 3. In the 'METHODOLOGY' section, the explanation of sequence length is somewhat disjointed and may not be easily understood by readers. 4. The description of the resolution DSW embedding is not very clear.

#significance:
The limited experimental results are a significant factor affecting its impact. If there were more comprehensive experimental results, this paper would have a significant impact. It has made a substantial contribution to the field by addressing the capture of long-term temporal dependencies in the time domain through an end-to-end model. While the preprocessing stage may pose some challenges, the practical utility of the model surpasses that of a simple machine learning classification model. It also avoids the need for complex feature engineering.

**Weaknesses:**

1. The originality of the model's architecture is somewhat limited. Despite its focus on temporal considerations, it relies on a frequency domain model, and its fundamental structure closely resembles that of FED-Former.

2. The model's real-world applicability raises questions. While it exhibits strong performance in just four tasks from a single competition, it lacks comprehensive comparisons with other existing models, making its effectiveness less persuasive. To bolster your argument, I recommend introducing additional datasets for comparative analysis, thereby reinforcing the model's practical utility.

**Questions:**

Questions:

In the methodology section, could you elaborate more on the sources and rationale behind your ideas, rather than merely describing the structure? In the EXPERIMENT SETTINGS section, could you provide specific details about your experiment's hyperparameters to facilitate model reproducibility for readers?

Suggestions:

To enhance the paper, start by conducting experiments on a broader spectrum of datasets to evaluate the model's performance and establish its feasibility. Make an effort to include comparative analyses with other existing models wherever applicable. Additionally, aim to incorporate more visual aids to offer a more lucid representation of your concepts. Given the numerous symbols within your paper, improving the intuitiveness of these notations is essential.


---------After rebuttal--------------
Thank the authors for providing the rebuttal. I have read the rebuttal and the other reviews. Based on the limited originality and the insufficient ablation studies as pointed out by other reviews, I decreased my rating to marginally below the acceptance threshold. Please consider the reviews and the rebuttal carefully for another submission.

---

> ### Author Response · Authors · 2023-11-21
> **Response to Reviewer zx9B**
>
> Thank you for your time and effort in reviewing our paper. We apperciate your through review of our paper and your insightful suggestion for enhancing our paper quality. The following are the response to your concern and question:
>
> > **S1: The representation of certain letters is not very intuitive and may lead to confusion.**
>
> Thank you for pointing this out. We will adjust them in our final version.
>
> > **S2: There is a lack of illustrations, making the descriptions less visually informative.**
>
> Thank you for your suggestion. We add prediction visualization in the appendix (in our current uploaded revised version) to better demostrate the rationale behind the module we proposed in this paper.
>
> > **S3:In the 'METHODOLOGY' section, the explanation of sequence length is somewhat disjointed and may not be easily understood by readers**
>
> Thank you for point this out. We revised the explaination of the sequence length in our currently uploaded revised version.
>
> > **S4: The description of the resolution DSW embedding is not very clear.**
>
> Thank you for point this out. In our final version, we will add a visualization for this to increase the clarity.
>
> > **W1: The originality of the model's architecture is somewhat limited. Despite its focus on temporal considerations, it relies on a frequency domain model, and its fundamental structure closely resembles that of FED-Former.**
>
> Thank you for point this out. We agree with your point that our model is smiliar with FEDformer in overall model architecture. The results of ablation study (please refer to Table1 in gerneral reponse) have proved the modification we made in our proposed model highly improve the model performance. We will revise our result section with the result of ablation study to emphasize the effectiveness of the module we proposed in our final version.
>
>
> >**W2: The model's real-world applicability raises questions. While it exhibits strong performance in just four tasks from a single competition, it lacks comprehensive comparisons with other existing models, making its effectiveness less persuasive. To bolster your argument, I recommend introducing additional datasets for comparative analysis, thereby reinforcing the model's practical utility.**
>
> Thank you for your suggestion. Due to most of the physiological signal emotion recognition dataset and benchmark are foucsing on the classification task, there are limited existing works that we can make the comparison. To make our results more persuasive, we add another transformer based solution which is also working on EPiC challenge dataset for comparison. We will also add our results on DECAF (https://decaf-dataset.github.io/DECAF/Description.html), which is also a physiological signal emotion recognition dataset with continous label, in our final version.
>
> >**Q1: In the methodology section, could you elaborate more on the sources and rationale behind your ideas, rather than merely describing the structure?**
>
> We add the discussion about the rationale of each component in the general response. We will revise our methodology section and move some of the formula into the appendix in our final version.
>
>
> >**Q2: In the EXPERIMENT SETTINGS section, could you provide specific details about your experiment's hyperparameters to facilitate model reproducibility for readers?**
>
> Thank you for point this out. We will make our codes open soure after the paper get accept.

---

> ### Author Response · Authors · 2023-11-23
> **Looking forward to your feedback**
>
> Dear reviewer zx9B,
>
> Thanks again for your valuable time and insightful comments. We are wondering if our response and revision have resolved your concerns. If our response has addressed your concerns, we would highly appreciate it if you could re-evaluate our work and consider raising the score.
>
> If you have any additional questions or suggestions, we would be happy to have further discussions.
>
> Best regards,
>
> authors

---

### Official Review · Reviewer_8PZe · 2023-10-31

**Soundness:** 3 good
**Presentation:** 2 fair
**Contribution:** 1 poor
**Rating:** 3
**Confidence:** 5

**Summary:**

This study proposes a new method to analyze the link between physiological signals and emotional changes. By using a transformer-based model, the authors shift the focus from emotion recognition to predicting sequences of multivariate time series data. They employ a two-stage attention mechanism to process these signals, and through multitask learning, improve the prediction of two emotional states: valence and arousal. The model, when tested on a specific dataset, outperformed other methods from a related challenge in all evaluated scenarios.

**Strengths:**

The authors achieved SOTA performance on the EPiC challenge dataset.

**Weaknesses:**

1. I am having trouble seeing the contributions of this paper. It seems like the authors have used an existing model and fitted it on a new dataset. The authors have not tested whether the model is generalizable to other affective datasets. If authors could point out some key aspects of the papers contributions a bit more clearly it would be greatly appreciated.
2. The interpretability of the results seem a bit lacking. Why does the author's proposed method outperform other participant's methods? Additionally, table 1 has a row labeled "Scenario level." Could the authors clarify what this means?

**Questions:**

1. Have the authors considered doing an ablation study on which physiological signals may be contributing the most to the performance?
2. The authors mention that they "capture the relationship between physiological signals and affective changes." Could you clarify to me how this relationship is captured with this model?
3. This paper was submitted to the "Primary Area: applications to neuroscience & cognitive science." I am having a bit of trouble seeing this paper's contribution to this area. Could the authors clarify this?

---

> ### Author Response · Authors · 2023-11-21
> **Response to Reviewer 8PZe**
>
> Thank you for your time and effort in reviewing our paper. The following are our responses to your specific question and Concern:
>
> > **W1: Contributions of this paper**
>
> 1. As Wavelet and Fourier transform are widely applied on physiological signal emotion recognition task for preprocessing. We want to investigate whether Wavelet and Fourier attentions have adavantages on physiological signal emotion recognition task. Our experiment resulst proves the Wavelet and Fourier attentions do have adavantages on predicting seasonal of target signal but has some limitation of predicition the trend of target signal. We address this problem via utilizing the time domain attention, such as Autocorrelation attention.
> 2. We redesigned the vanilla Frequency domain attention and Time domain attention by applying Two-Stage Attention (TSA) on them. Therefore, the refering frequency domain attention and time domain attention are not only calculating on the time-step wise but also on channel wise. Our experiments proves the effectiveness of the proposed attention.
> 3.  The tranditional time series prediction task is working domain consistent data, where the input and target data are in the same domain. However, for physiological signal emotion recognition task, the input and target signals has different domain and sample rate, which turns physiological signal emotion recognition task to be relatively more challenging. In this paper, we successfully apply the SOTA methods from multivariate time series prediction task to physiological signal emotion recognition task. Moreover, the proposed method achieves SOTA result in physiological signal emotion recognition task. We believe our solution and results could inspire the future research in the field.
>
> >**W2: Why does the author's proposed method outperform other participant's methods?**
>
> The major resons are:
>
> 1. The refered technique individually has been proved their effectiveness in time series prediction tasks and achieves SOTA results when published.
> 2. We redesign the refered methods to make them can be applied to emotion recognition task.
> 3. The wavelet and fourier transform technique utilized in frequency domain attention has been widely applied in physiological signal emotion recognition task and proves their effectiveness.
>
> >**Q1: Have the authors considered doing an ablation study on which physiological signals may be contributing the most to the performance?**
>
> Thank you for point this out. We will add this ablation study in the final version of our paper.
>
> >**Q2: The authors mention that they "capture the relationship between physiological signals and affective changes." Could you clarify to me how this relationship is captured with this model?**
>
> The details of this process are the following:
> 1. The physiological signal input will be first convert to a time-frequency domain representation through the encoder.
> 2. The decoders' self-attention blocks will extract the frequency domain representation and time domain representation from the affective signal projection.
> 3. The decoders' cross attention block will capture the frequency domain dependencies and time domain dependencies between the physiological signal and the affective signal projection via the extracted representaions.
>
> >**Q3: This paper's contribution to "Primary Area: applications to neuroscience & cognitive science."**
>
> 1. We explore and examine the SOTA methods from time series prediction task on physiological signal emotion recognition task.
> 2. We redesign the refering methods and propose a new solution for physiological signal emotion recognition task.
> 3. The proposed solution achieves SOTA reuslt in physiological signal emotion recognition task.
> 4. We believe our results and solution could inspire the future reseach in the field.

---

> ### Author Response · Authors · 2023-11-23
> **Looking forward to your feedback**
>
> Dear reviewer 8PZe,
>
> Thanks again for your valuable time and insightful comments. We are wondering if our response and revision have resolved your concerns. If our response has addressed your concerns, we would highly appreciate it if you could re-evaluate our work and consider raising the score.
>
> If you have any additional questions or suggestions, we would be happy to have further discussions.
>
> Best regards,
>
> authors

---

### Official Review · Reviewer_s9qV · 2023-10-31

**Soundness:** 2 fair
**Presentation:** 1 poor
**Contribution:** 1 poor
**Rating:** 3
**Confidence:** 5

**Summary:**

In this paper, the authors propose a transformer-based solution for physiological signal emotion recognition, by converting the recognition task to a multivariate time prediction task. The authors decompose the signals into separate time and frequency domain representations using self-attention mechanisms and capture channel dependencies. The proposed system outperforms other participants in the Emotion Physiology and Experience Collaboration challenge.

**Strengths:**

1. The authors utilizes some state-of-the-art mechanisms, such as wavelet and Fourier attention, to decompose the physiological signals into separate frequency domain and time domain representations.
2. The authors introduce a two-stage attention mechanism to capture the dependencies between signals. This addresses the challenge of having both cross-time and cross-dimension dependencies and improves the model's ability.

**Weaknesses:**

1. There should be more ablation studies to prove each component in the proposed method works.
2. The related work section introduces some references and attention mechanisms that the authors used, without the information about the following compared works.
3. The replace of the MSAs in the encoder should have evidence to prove its rationality and efficiency, so as the frequency and time domain TSA in the decoders.
4. Many simple formula notations are cumbersome, for example, we have long been familiar with the attention mechanisms.

**Questions:**

Please refer to the weaknesses.

---

> ### Author Response · Authors · 2023-11-21
> **Response to Reviewer s9qV**
>
> Thank you for your time and effort in reviewing our paper and insightful suggestion. The following are our response to your concerns:
> >**W1: There should be more ablation studies to prove each component in the proposed method works.**
>
> Thank you for point this out. We add the result of ablation study in the general response. We will revise our result section in our final version.
>
> >**W2: The related work section introduces some references and attention mechanisms that the authors used, without the information about the following compared works.**
>
> The refered model and attention mechanism are mostly designed for time series prediction, where the input and target data are in the same domain. For physiological signal emotion recognition, the input and target signal are in different domain and have different sample rate. The models proposed for time series prediction couldn't be directly applied on physiological signal emotion recognition. One contribution of our paper is that we proposed a solution for applying the SOTA time series prediction model on physiological signal emotion recognition, which we believe could inspire future research in the field.
>
> >**W3: The replace of the MSAs in the encoder should have evidence to prove its rationality and efficiency, so as the frequency and time domain TSA in the decoders.**
>
> Thank you for point this out. We add the explaination our rationale for each component and add the result of ablation study to prove our rationale and efficiency. We will revise our methodology section and result section accordingly in our final version.
>
> >**W4: Many simple formula notations are cumbersome, for example, we have long been familiar with the attention mechanisms.**
>
> Thank you for point this out. We will revise our methodology section and fix the notation problems in our final version.

---

> ### Author Response · Authors · 2023-11-23
> **Looking forward to your feedback**
>
> Dear reviewer s9qV,
>
> Thanks again for your valuable time and insightful comments. We are wondering if our response and revision have resolved your concerns. If our response has addressed your concerns, we would highly appreciate it if you could re-evaluate our work and consider raising the score.
>
> If you have any additional questions or suggestions, we would be happy to have further discussions.
>
> Best regards,
>
> authors

---

### Author Response · Authors · 2023-11-21
**General Response**

We thank all the reviewers for their elaborate and constructive feedback. The following are our response to common concerns of all the reviwers:

**Ablation Study**

| Module | RMSE | CCC |
| :------: | ---- | --- |
| Time Decoder + Frequency Decoder | 0.9370 | **0.0352** |
| TSA + Time Decoder | 0.8576 | 0.0290 |
| TSA + Frequency Decoder | **0.7582** | 0.0198 |
| TSA + Time Decoder + Frequency Decoder* | 0.7650 | 0.0288 |

Table 1 presents the result of ablation study on Scenario 1. The evaluation metrics are root mean square error (RMSE) and concordance correlation coefficient (CCC), which is range from -1 to 1 indicating negative correlation and postive correlation respectively. The evaluation metrics in the table take average results of valence and arousal. **Bold** presents the best results of relative metrics.

1* The result of Scenario 1 has been updated. The previous result was 0.8288 (average RMSE of valence and arousal). We revise the formula of the final output.

**Analysis of the Ablation study**
We use RMSE and CCC as the metrics to reflect the model's ability on predicting seasonal and trend. Without two stage attention(TSA), the model with Time Decoder and Frequency Decoder has advantage on prediction trend and disadvantage on predicting seasonal. While the model with TSA, has advantage on predicting seasonal and disadvantage on trend. Our proposed method has the best tradeoff between the RMSE and CCC. We add predicition visualization in the appendix A.6 to better demostrate this. More explainations of the rationale and effectiveness of each component are in the following secton.


**Rationale of Frequency Domain Attention**

Wavelet Transform and Fourier Transform has been widely applied in the physiological signal emotion recognition task as preprocessing method. As Frequency attention utilizes Wavelet and Fourier Transform technique and self-attention mechanism, we believe it has advantages on encoding the input physiological signal, decoding the frequency domain information of the target affective signal and capture the frequency domain relationship between the physilogical signal and affective signal.

**Rationale of Time Domain Attention**

While Frequency attention works fine on mitigate the seasonal difference between the prediction and target. It is not good at capturing the trend difference between the predicion and target on time domain. From Table 1, model with TSA and Frequency decoder has the best performance according to the MSE. However, model with TSA and Time decoder has best performance according to CCC among the models add TSA. To combine the advantage of both type decoder, we add them together. The model with TSA and both decoders achieves comparable performance to the models with individual decoder.

**Rationale of Two Stage Attention**

The tranditional transformer based solution for time series task applied attention only on time step wise for each channel, where the channel wise dependencies are neglected. In our task specifically, TSA can provide the dependencies among each physiological signal and each affective signal addition to the vanilla Frequency and Time domain attention. From Table 1, adding TSA improve the model performance by 24.85% according to RMSE.

**Revision**
The following are the revsion we made in the currenly uploaded version:
1. We add prediction visualization in the appendix to better demostrate our rationale for each component.
2. We add one more transformer based solution in the result section for more comprehensive comparison
3. The final output formula in section 3.2.3 has been revised.

---

### Meta-Review · Area_Chair_um7N · 2023-12-06

**Metareview:**

In this paper, the authors propose a transformer-based solution for physiological signal emotion recognition, by converting the recognition task to a multivariate time prediction task. The authors decompose the signals into separate time and frequency domain representations using self-attention mechanisms and capture channel dependencies. The proposed system outperforms other participants in the Emotion Physiology and Experience Collaboration challenge.

Strengths:

-The authors utilizes some state-of-the-art mechanisms, such as wavelet and Fourier attention, to decompose the physiological signals into separate frequency domain and time domain representations.
-The authors introduce a two-stage attention mechanism to capture the dependencies between signals. This addresses the challenge of having both cross-time and cross-dimension dependencies and improves the model's ability.

Weaknesses:
- Contributions and results seem lacking
- It is not clear that the model is generalizable as it is only evaluated on one dataset.
-There should be more ablation studies to prove each component in the proposed method works.
-The related work section introduces some references and attention mechanisms that the authors used, without the information about the following compared works.
-The replace of the MSAs in the encoder should have evidence to prove its rationality and efficiency, so as the frequency and time domain TSA in the decoders.
-Many simple formula notations are cumbersome, for example, we have long been familiar with the attention mechanisms.
-The method is only evaluated on one dataset

**Justification For Why Not Higher Score:**

The paper is not strong enough to be considered competitive for ICLR.  The results are not foundationally novel, but rather a novel application of known methods to a particular problem.  It is not clear that the pipeline works well on any other dataset besides the one tested.

**Justification For Why Not Lower Score:**

The paper is a good application paper, but it is highly specific not just to one area but to one dataset  and would probably receive more recognition at a conference like ACII (the dataset that they used was developed as a challenge set for an ACII workshop).  As that conference will not be held again for a while the authors could submit to the IEEE Transactions on Affective Computing, the journal associated with the conference.

---

### Decision · Program_Chairs · 2024-01-16

Reject